# Performance characterization of a laminar gas-inlet

Da Yang[1,2,3,*], Margarita Reza[1,2], Roy Mauldin[4], Rainer Volkamer[1,2,4], and Suresh Dhaniyala[3,*]

[1]Department of Chemistry, University of Colorado Boulder, Boulder, CO
[2]Cooperative Institute for Research in Environmental Sciences (CIRES), University of Colorado Boulder, Boulder, CO
[3]Mechanical and Aeronautical Engineering, Clarkson University, Potsdam, NY
[4]Dept of Atmospheric and Oceanic Sciences, University of Colorado Boulder, Boulder, CO

*Correspondence to: Da Yang (da.yang@colorado.edu); Suresh Dhaniyala (sdhaniya@clarkson.edu)

**Abstract.** Aircraft-based measurements enable large-scale characterization of gas-phase atmospheric composition, but these measurements are complicated by the challenges of sampling from high-speed flow. Under such sampling conditions, the sample flow will likely experience turbulence, accelerating and mixing of potential contamination of the gas-phase from the condensed-phase components on walls and reduced vapor transmission due to losses to the inner walls of the sampling line. While a significant amount of research has gone into understanding aerosol sampling efficiency for aircraft inlets, a similar research investment has not been made for gas sampling. Here, we analyze the performance of a forward-facing laminar flow gas inlet to establish its performance as a function of operating conditions, including ambient pressure, freestream velocities, and sampling conditions. Using computational fluid dynamics (CFD) modeling we simulate flow inside and outside the inlet to determine the extent of freestream turbulent interaction with the sample flow and its implication for gas sample transport. The CFD results of flow features in the inlet are compared against measurements of air speed and turbulent intensity from full-sized high-speed wind-tunnel experiments. These comparisons suggest that the Reynolds Averaged Navier-Stokes (RANS) CFD simulations using the Shear Stress Transport (SST) modeling approach provide the most reasonable prediction of the turbulence characteristics of the inlet.

## 1 Introduction

Atmospheric characterization for climate studies requires complete knowledge of both the gas-phase and aerosol constituents. While some of this characterization can be done from ground and satellite stations, aircraft-based measurements are critical for validation of remote measurements and for capturing variations in the atmospheric concentrations of trace species at fine spatio-temporal scales, and with vertical resolution. Such measurement capabilities are critical for improving our understanding of the physical and chemical processes in the upper troposphere/lower stratosphere (UTLS) (Volkamer et al., 2015; Wang et al., 2015; Koenig et al., 2017, 2020; Brenninkmeijer et al., 1999; Karion et al., 2010; Filges et al., 2015). While aircraft measurements in the open atmosphere are possible, and have advantages for detecting radical species without complications from inlet lines (Volkamer et al., 2015; Wang et al., 2015; Koenig et al., 2017, 2020), most aircraft

measurements rely on capturing samples from the ambient under high flow speed using inlets. The samples are then transferred and analysed for aerosol/gas concentrations and compositions using sensitive instruments inside the aircraft. Accurate measurements require well-designed and characterized sampling systems. Such sampling systems are now relatively common

for general aerosol measurements (Kulkarni et al., 2011), for specifically sampling aerosol in clouds (Moharreri et al., 2014, 2013), capturing aerosol particles without contamination from the gas-phase constituents (Dhaniyala et al., 2003), etc, and widely deployed in field studies. For gas-phase measurements, however, there are very limited number of studies describing and characterizing aircraft sampling systems.

When gas-phase species being analysed also exist in the condensed phase, separation of the two phases is necessary. Some gas-phase inlet designs that have addressed this need include the rear-facing inlet (e.g., Kondo et al. 1997) and downward facing inlet (Fahey et al., 1989). When gas-particle separation is unimportant, a simpler forward-facing gas inlet design (e.g. Ryerson et al. 1999; Dhaniyala et al. 2003) can be used. While the different inlet designs have been deployed in field campaigns, their performance has rarely been fully characterized and hence gas-species transport efficiency of these inlets is

largely unknown. This lack of information on transport efficiency results in significant uncertainty in quantitative measurements made by these inlets.

Gas-transport efficiency is a strong function of the flow field within the inlet and the sampling tube material. While bench-top experiments can capture the interaction of tubing material with different gas-species, understanding the role of the flow

field in gas-species transport requires field experiments and modelling studies. The combination of high freestream speeds upstream of the inlet and low sample speeds in the inlet result in turbulent flows at the inlet entrance. Considering this entrance turbulence, understanding the relative merits of slowing sample flow speeds for laminar flow and minimizing residence time is critical for optimizing the sample tube design for efficient gas-sample transport under different aircraft conditions.

In this study, we use computational fluid dynamics (CFD) simulations to characterize gas-sampling efficiency of a forward-facing inlet as a function of aircraft operating conditions. Section 2 describes the gas inlet, which is being developed and certified within the framework of the TI$^3$GER (Technological Innovation Into Iodine for GV aircraft Environmental Research) field campaign, and over the long term makes a new form factor inlet available for wider use by the atmospheric research community. Section 3 describes the model simulations to predict the internal flow response and provide windtunnel

experiments to evaluate and optimize the CFD model. Section 4 discusses atmospheric applications of the inlet over a range of sampling conditions. While simulations of the expected gas-sampling efficiencies obtained from this study are specific to the inlet design studied here, the impact of turbulence on species loss will inform the operational space of inlets of all designs. Finally, we summarize our findings in section 5.

## 2 Methods

 ### 2.1 Description of the laminar flow inlet

The laminar flow inlet is based on the design described by Eisele et al.(Eisele et al., 1997), which has flown previously on an NSF twin otter and the NASA DC-8 aircrafts. This inlet has proven to be superior in straightening and slowing sample flow, while allowing effectively "walless" sampling. The modified design described here is shown in Figure. 1. The inlet consists of a shroud, outer inlet tube, with an inner inlet tube and a sample tube nested inside. Using elliptical cross-sections for the leading edge of the shroud and outer inlet tube allows flow straightening without separation and is tested here over a range of angle of attacks (Eisele et al., 1997). The use of rear restrictions allows for controlled multistage flow slowing. The design provides for in-situ calibration of OH and $H_2SO_4$ (Mauldin III et al., 1998), and other strong acids, e.g., iodic acid (Finkenzeller et al., 2023). The calibration is accomplished by producing a known amount of OH in front of the sampling inlet by photolyzing ambient $H_2O$ present in the sampled air with the 184.9 nm emission line from a filtered Hg Pen-Ray lamp. Downstream, OH reacts with $SO_2$ added through a pair of injectors inside the sample tube, producing $H_2SO_4$ which is then detected via nitrate ToF-CIMS (Eisele and Tanner, 1991). This study goes beyond previous work by shrinking the inlet size, developing a CFD model of the laminar gas inlet, and examining the flow characteristics using measurements of velocity and turbulent intensity inside the inlet to evaluate the CFD model. The chemical aspects of calibration are beyond the scope of this paper. In the current study, the port used to fit the Pen-Ray lamp is used to fit a hot wire probe in location H, as is illustrated in Figure 1. Additional hot wire measurements are made inside the sampling tube to characterize flow velocity and flow turbulent intensity (locations 2" and 3" in Fig. 1). The sample flow velocity is further measured using a Pitot tube inside the inlet (location P in Fig. 1). Our focus here is on design aspects of the CFD model to optimize the restrictor size in the inlet design, in order to minimize turbulent intensity inside the inlet, and assess inlet performance over the range of operating conditions expected for the Gulfstream 5 aircraft.

The inlet consists of two shrouds to slow and straighten the flow prior to the sampling inlet. The outer shroud is a cylindrical tube of diameter ~ 3 inch (7.62 cm), that acts to align the flow axially, independent of the aircraft angle of attack. Inside the outer shroud, starting ~ 6 inch (15.24 cm) downstream of the leading edge, is a 2 inch (5.08 cm) diameter cylinder, as shown in Figure 1. This inner shroud acts to sub-sample from the core flow of the outer shroud, eliminating flow that might have had contact with the outer shroud walls. The inner and outer shroud leading edges have a blunt airfoil shape to minimize flow separation. Within the inner shroud, a sampling tube of diameter 0.75 inches (1.9cm) is located ~ 5 inch (12.7cm) downstream of the shroud's leading edge. After a 90° bend, the diameter of the sampling tube is reduced to 0.5 inch (1.27cm) prior to passing through the aircraft hull.

For maximal gas transport efficiency, it is typically assumed that maintaining a laminar flow in the sampling tube results in the lowest wall losses. To ensure flow laminarization in the sample tube is achieved over a short distance, it is critical that the

flow enters the sample tube with minimal turbulence. It is expected that turbulence in the sample tube will be generated when the flow is suddenly decelerated upon entering the tube. Thus, ideal sampling conditions will require the flow velocity in the sample tube to be reasonably matched with that just upstream of the sample tube. This can be done by appropriately selecting the inner and outer shroud geometry, particularly the exit section of the shrouds. A desired sampling flow of ~ 10 to 40 LPM in the inlet corresponds to a sample velocity of ~ 1 to 5 m s$^{-1}$ at ground level atmospheric conditions. To obtain a near-isokinetic sampling condition under a cruise speed of 0.75 Mach for the GV aircraft, the flow velocity just upstream of the sample tube must be reduced by a factor of at least 40. This sudden reduction can result in significant turbulence. To minimize turbulence generation, flow reduction over several stages can be considered. This multi-step reduction in velocity can be achieved by constricting and controlling the flow into the two shrouds, such that each step decreases the velocity by a factor of 2-4. The impact of this multi-step velocity reduction on the flow characteristics in the sample tube must be fully understood for determining its gas sampling efficiency.

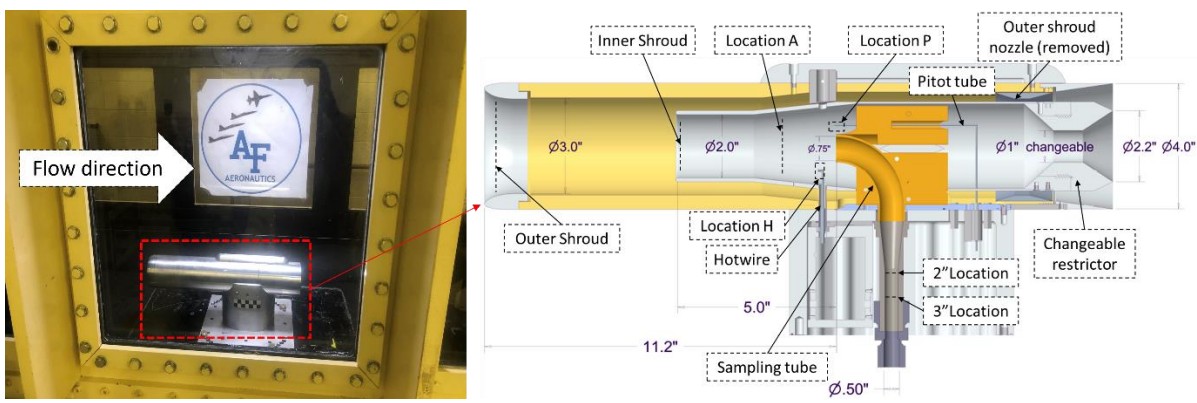

**Figure 1: (a) The inlet located in the wind tunnel test section; (b) Cross-section side view of the laminar gas inlet showing the different flow regions and probe locations: (Location P) pitot tube; (Location H) hot wire; (2" location) and (3" location) mark the locations of hot wire measurements inside the sampling tube.**

### 2.2 Computer fluid dynamics (CFD) model simulations

To determine the internal inlet flow characteristics, computational fluid dynamics (CFD) simulations are conducted in and around the inlet for a range of conditions. The aircraft operating conditions of high-speed and variable pressure and temperature necessitates considering compressible effects on the inlet airflow. Additionally, as the Reynolds number of the flow in and around the sampling inlet is high, the modelling must account for turbulence in the flow. Considering the very different flow conditions in the different regions of the inlet, the selection of the most optimal turbulence model for Reynolds-Averaged Navier Stokes (RANS) simulations is not obvious. Here, we consider two different turbulence models: 2-equation realizable k-ε and transition SST models, and determine their predicted average-flow conditions and turbulence intensities. The standard k-ε model considers turbulent kinetic energy (k) derived from flow fluctuation velocity and its dissipation rate (ε) and has been previously used extensively for modelling external flow around aircraft inlets (Craig, et al. 2013; Craig, et al.

2013). The realizable k-ε turbulence model is more reliable for flows in complex regions with separated boundary layers, which will be important inside shrouds (Shih et al., 1995). For modelling flow in the sub-sample tube, as the flow transitions

from turbulent flow in the shroud to laminar flow in the tube, the transition SST k-ω model might be most relevant (Menter et al., 2004).

Here, we use the commercial code FLUENT 18.1 (ANSYS, NH) for the inlet simulations. To calculate the flow field around the inlet a large rectangular domain around the inlet was chosen. On one side of the external domain is the aircraft hull onto

which the inlet is installed, and this surface is set to a wall boundary condition. The other boundaries represent the freestream, and are set to pressure-far-field, with pressure, temperature, and Mach number values selected to match flight conditions. The simulations were modelled assuming steady state flow.

Domain and mesh size insensitivity tests were conducted for the case of 220 m s$^{-1}$ freestream velocity and 15,000 Pa ambient

pressure, inner shroud restrictor size of 25mm, and an exit sample flow velocity of 2.4 m s$^{-1}$. To establish the ideal domain size, we studied several sizes of external modelling geometry around the inlet and for each case, we probed and compared velocity at the entrance to the outer shroud. This series of tests established an optimal external domain size with length, width, and height as 11.7*Ht, 6.7*Ht, and 5*Ht, where Ht is the height of the inlet assembly from the aircraft hull to the inlet's top edge (~30 in cm). Increasing the domain beyond this size did not result in any change in the monitored velocity at the entrance

to the outer shroud. A similar test with varying number of meshes in the regions of strong velocity and pressure gradients showed that a final mesh with ~ 3 million cells resulted in less than a 1% change in velocity at the outer shroud entrance.

**2.3 Boundary conditions of flow simulations**

We conducted our CFD simulations for a range of conditions consistent with ground-level wind-tunnel experiments and high-altitude flights (Table 1). The simulations for each case were run until convergence for mass (flow and gas species),

momentum, energy, and turbulence parameters. The critical criterion for all our simulations was mass convergence, and for all cases, we ensured that the residual for this parameter decreased to less than 5e-3 and did not change with further increase in the number of iterations.

The ground level boundary conditions, shown in Table 1, are used for comparing k-ε model and SST model. The predictions

of CFD-calculated inlet performance under different design conditions were evaluated against high-speed wind-tunnel experiments for a selected set of operating conditions. In addition, we conducted CFD simulations for high-altitude freestream conditions of 0.75 Mach and 15,000 Pa, and a range of restrictor sizes, as listed in Table 1. For the high-altitude cases, simulations were only conducted with the SST turbulence model.

**Table 1. Boundary conditions for CFD simulations under wind-tunnel and high altitude conditions**

| Variables | Free stream conditions | |
| --- | --- | --- |
| | Ground (Wind-tunnel) | High-altitude (aircraft) |
| Static pressure (Pa) | 97000 | 15000 |
| Static temperature (K) | 285 | 220 |
| Size of restrictor (mm) | 25, 17, 12.5 | 25, 17, 12.5, 10, 6.25 |
| Freestream velocity (m s$^{-1}$) | 180, 145, 102, 75 | 220 |
| Freestream turbulent intensity | 3%, 1%, 0.5% | 3% |
| Angle of attack (°) | 3, 20 | 3 |
| Sampling flow rate (m s$^{-1}$) | 2.4 | |

## 2.4 Wind tunnel experiments

Experimental tests to determine the velocity in the inner shroud and turbulence characteristics within the inlet were made in the Air Force Academy's high-speed tunnel (Colorado Springs, CO). The recirculating wind tunnel has a 1 ft by 1 ft test

section, with a maximum flow velocity of 180 m s$^{-1}$ (Fig. 1a). The wind-tunnel is fitted with a pitot tube to measure freestream velocity and temperature and pressure sensor to measure freestream air properties. The flow conditions and inlet operating conditions relevant for wind-tunnel tests are listed in Table 2.

**Table 2. Restrictor sizes and operating conditions used in the wind-tunnel experiments.**

| Variables | Ground level wind tunnel measurement | | |
| --- | --- | --- | --- |
| Size of Restrictor (mm) | 12.5 | 17 | 25 |
| Angle of attack (°) | 0 | 0, 10, 20 | |
| Freestream flow velocity in (m s$^{-1}$) | 75, 130, 180 | | |


A test inlet was fabricated in the University of Colorado's machine shop facility. The inlet was installed on the floor of the tunnel on a rotating plate to vary the inlet angle of attack. Inside the inlet, a pitot tube assembly is incorporated just above the sampling tube entrance (shown as location P in Fig. 1b) in the inner shroud to monitor the average flow velocity just upstream of the sample tube entrance. The pressure measurements from the pitot tube were made at 20Hz. A hotwire probe (Dantec

Dynamics Miniature wire probe 55P13 with 55H20 probe support) that has a velocity measurement range of 0.2 to 500 m s$^{-1}$ was used to capture turbulence intensities in the inlet. The hotwire was located upstream of the sample tube (shown as location

H in Fig. 1b) for measurements of turbulence characteristics in the inner shroud. The hot-wire measurements were made at 1 kHz. The wind-tunnel measurements were made for freestream velocities ranging from 30 m s$^{-1}$ to 180 m s$^{-1}$ (Table 2). A hotwire probe (Dantec Dynamics Miniature wire probe 55P11 with 55H21 probe supports) was placed at 2" and 3" locations
within the sampling tube (see Fig. 1b) to measure turbulent intensities there.

## 3 Results

### 3.1 Shroud flow

The CFD simulations of flow field in and around the inlet were conducted under both wind-tunnel and aircraft conditions listed in Table 1. The first simulations were conducted with the 25 mm restrictor under wind-tunnel conditions so as to understand
the prediction differences of the two turbulence models. The velocity contour field for a wind-tunnel speed of 180 m s$^{-1}$ at ground level is shown in Figure 2a. The presence of the pylon creates some lack of symmetry in the external flow field and also in the aft-end of the interior flow field. With the significant pressure drop provided by the constriction in the exit sections of the outer and inner shrouds, however, near uniform flow is achieved across the inner and outer shroud cross-sections. Also, the high pressure drop in the narrow exit channels results in exit velocities close to Mach 1 (see Fig. 2a), ensuring that the flow
inside the inlet will be largely invariant with small changes in ambient velocities.

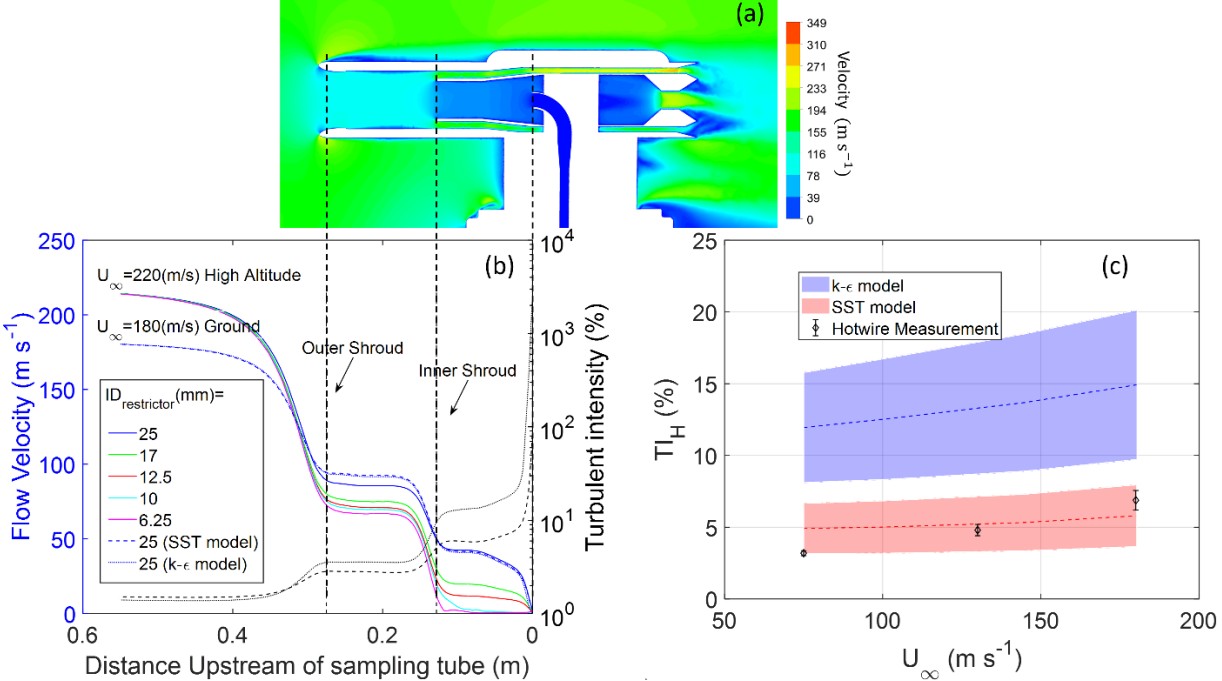

Figure 2: (a) Contour plot of velocity field in and around the inlet with 25mm restrictor for a freestream velocity of 180 m s$^{-1}$, freestream turbulence intensity of 3%, and angle of attack of 3°. (b) Centerline velocity for different restrictor sizes for aircraft (high altitude) and wind-tunnel (ground) conditions. Also, shown are the turbulence intensities for the k-ε and SST models for the case of
25mm restrictor and 180 m s$^{-1}$ freestream velocity. (c) The turbulent intensities calculated at location H using the k-ε and SST models

for different wind-tunnel velocities ($U_\infty$) compared to wind tunnel measurements. The shaded areas for each plot represent the range of turbulent intensities obtained from simulations for freestream turbulent intensities varied from 0.5% to 3% (see Fig. S1 for additional details). The error bar for experimental data is the standard deviation of repeated wind tunnel measurements.

The velocity and flow turbulence variation along the centerline of the inlet as the flow moves from the freestream to the entrance of the sampling tube inlet is shown in Figure 2b. For a freestream velocity of 180 m s$^{-1}$, a sample velocity of 2.4 m s$^{-1}$, and under wind-tunnel conditions, the 25 mm restrictor results in a velocity reduction factor of ~ 2.2 in each of the two shrouds. As the flow reaches the sample tube, the velocity adjusts to the value required to maintain the sampling flowrate (set to 2.4 m s$^{-1}$ for this case). The velocity drop from the inner shroud velocity of 40 m s$^{-1}$ to the sampling tube velocity happens dramatically over a short distance just upstream of the tube entrance. Reducing the inner shroud restrictor size reduces flow velocity in both shrouds with a larger decrease in the inner shroud. The prediction of average velocity along the inlet centerline is seen to be largely independent of the turbulence model used (within 2% of each other). This is as expected, as both the k-ε and SST models solve the same RANS equations.

Along with changes in velocity, the turbulence intensity changes as the flow moves from freestream to inside the inlet. For the wind-tunnel case of 180 m s$^{-1}$, both the RANS turbulence models predict a similar trend of increase in turbulence intensity with slowing flow, with the turbulence intensity increasing from the freestream value of 3% to over 100% at the sample tube entrance (Fig. 2b). The two models, however, predict different magnitudes of turbulence intensity, with the predictions of the realizable k-ε model being about twice that of the transition k-ω SST model. As gas transport characteristics under turbulent conditions are strongly dependent on turbulent diffusivity, it is important to accurately model turbulent intensities in the flow. The very different prediction of the two models needs to be resolved through wind-tunnel tests.

A critical characteristic of turbulence at any location is the fluctuation velocity ($u'$) at that location. This is calculated from the standard deviation of the measured velocity ($u_i$), as $u' = \sqrt{\frac{1}{N-1}\sum_{i=1}^{N}(u_i - \bar{u})^2}$, where $\bar{u}$ is the average flow velocity. The turbulent intensity (I) is then calculated as the ratio of fluctuation velocity to the average velocity, i.e. $I = \frac{u'}{\bar{u}}$. As the Pitot tube measurement frequency of ~20Hz is inadequate for fully characterizing the fluctuating velocity component was calculated only from the hotwire measurements. A comparison between CFD results of turbulent intensities in the inner shroud (location H) from the two different turbulence models studied and the wind tunnel experiments are shown in Figure 2c. Experimentally determined from velocity measurements made using pitot tubes in the wind tunnel freestream flow, the CFD results are shown as a shaded band covering the range of inner shroud turbulence intensities obtained for freestream turbulence ranging from 0.5% to 3%. Note that the turbulent intensity from CFD results is calculated by using the magnitude of velocity fluctuation divide the magnitude of velocity, $I = \frac{\sqrt{(u'^2+v'^2+w'^2)}}{\sqrt{(u^2+v^2+w^2)}} = \frac{\sqrt{(2k)}}{\sqrt{(u^2+v^2+w^2)}}$, where k is turbulent kinetic energy predicted from the model. Our hotwire measurement data suggests that the experimental data of turbulence intensities reasonably match

predictions of the SST model and differs significantly from that of the k-ε model. A closer look at the experimental data suggests that the match of the numerical predictions with measurements requires considering higher freestream turbulence with increasing freestream velocities. This is consistent with our observation that freestream velocities measured with a pitot tube were increasingly variable or noisy with increasing freestream velocities. This comparison result suggests that transition SST model is more accurate for our simulations of turbulence in the shrouds.

In addition to validating the turbulence model, wind tunnel experiments were also conducted to determine the relation between restrictor size and inner shroud velocities. A comparison of the experimental data obtained using a pitot tube at location P (Fig. 1) with CFD results at the same location, for varying freestream velocities is shown in Figure 3a. For this comparison, the CFD simulations at wind-tunnel conditions were repeated for two additional restrictor sizes other than 25 mm: 12.5 mm and 17 mm. For comparison of CFD predictions with experimental data, numerical results of flow velocity and turbulence are extracted and averaged over a small area of $1*0.5$ cm$^2$ around the measurement location. This averaging over the selected area provides a fair comparison in a location of strong gradients in flow properties. As the predictions of average velocity by the k-ε and SST models are nearly identical, we only show results for the SST case. A strong gradient in the CFD results of flow velocities at the pitot tube location results in a broad prediction curve at the different test conditions. In the comparison plot, the uncertainty of the experimental data represents the standard deviation of velocities obtained for a selected operating condition. The large uncertainties result from variation in freestream conditions over several days of experimentation and fluctuations in pitot tube pressures.

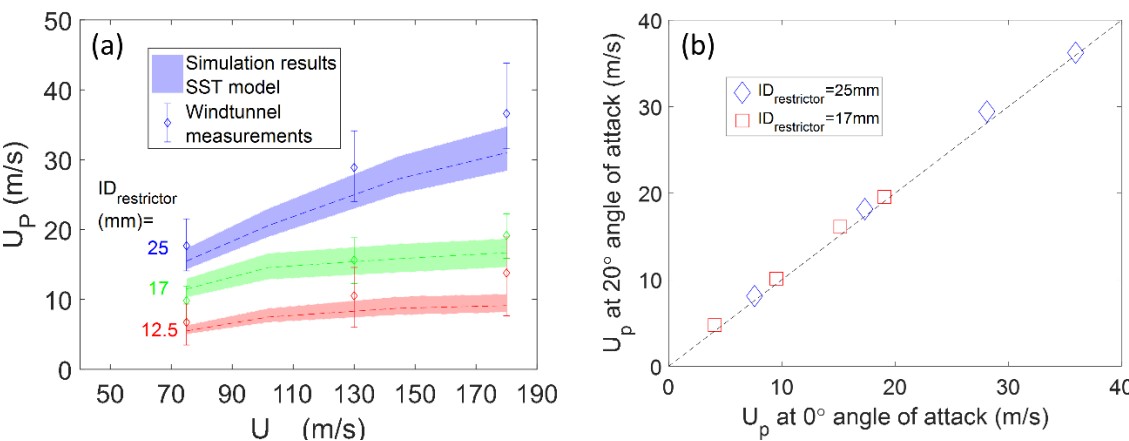

**Figure 3: Comparison of flow velocity measurements in the inner shroud (location P) with (a) simulations from the SST model as a function of freestream velocity, and (b) experiments of 25mm and 17mm restrictor as a function of angle of attack. The experimental data is the average value from all repeated wind tunnel measurements under the same operating conditions; the experimental error is obtained as the range of standard deviation from repeat measurements. Shading represents the range of simulation results of velocity over an area of $1*0.5$cm$^2$ at location P for varying TI$_\infty$ from 0.5% to 3%.**

The comparison plot shows that there is reasonable agreement of measured data with simulations for average velocities at the pitot-tube location over the entire range of freestream conditions, providing initial validation of our calculation approach. The wind tunnel measurement of inner shroud flow velocity is seen to be slightly (~ 10-20%) higher than the simulation results for the three restrictors and different freestream conditions tested. This difference could be because of a combination of a large spatial gradient in flow velocities at the measurement location, small errors in aligning the pitot tube during experiments, and incorrect drawing of the exact geometry of the pitot tube in the numerical model.

In Figure 3b, wind-tunnel results showing comparison of velocities measured by the Pitot tube at two different angles of attack for 4 wind-tunnel freestream conditions is shown. At all wind-speeds, the flow velocity at location P is independent of angle of attack, suggesting that the shrouds straighten flow as predicted by CFD simulations (Fig. S2).

**3.2 Sample tube flow**

The flow from the inner shroud enters the sampling tube at a rate dependent on the sampling boundary condition at the exit of the tube. The turbulence characteristics in the tube are critical for determining the fate of the sampled gas to the sampling instruments. The turbulence in the flow just upstream of the sampling tube and sudden reduction in velocity can generate turbulence in the entrance of the sampling tube that will not immediately dampen out. Additionally, the 90° bend immediately downstream of the entrance region, necessary to turn the flow into the aircraft, can create secondary flow that will also delay flow laminarization. Downstream of the 90° bend, the tube contracts from 0.75 inch (1.9cm) diameter to 0.5 inch (1.27) diameter. This contraction should help dampen any upstream turbulence.

The CFD results can help us understanding the developing length for the flow to reach near laminar flow. We examined flow in the sample tube for an average sampling flowrate of ~19 SLPM, which results in an exit velocity of ~2.4 m s$^{-1}$ in the sampling tube. This results in a Reynolds number in the sampling tube ~ 2300~2400. The surface-weighted average of turbulent intensities along the length of the sampling tube are shown in Figures 4a and 4b for the 25mm restrictor with 180 m s$^{-1}$ freestream velocity. Both the models predict that the turbulence dramatically rises from the freestream value at the entrance and then dampens as the flow develops along the length of the sampling tube. There are relatively minor differences in the decay patterns. In Figure 4c, the turbulence intensities at two locations within the inlet, 2" and 3" after the contraction of the sample tube (Fig. 1), are shown as a function of freestream velocities. These locations were chosen because they were experimentally easy to probe with a hot-wire. At each freestream velocity, the freestream turbulence intensity was varied from 0.5 to 3%. The resulted range of turbulent intensities obtained at any freestream velocity is represented as a shaded region. It is observed that the SST model predicts an increasing turbulence intensity with increasing freestream velocities and freestream turbulence, while the turbulence intensities predicted by the k-ε model are relatively insensitive to freestream velocity or freestream turbulence.

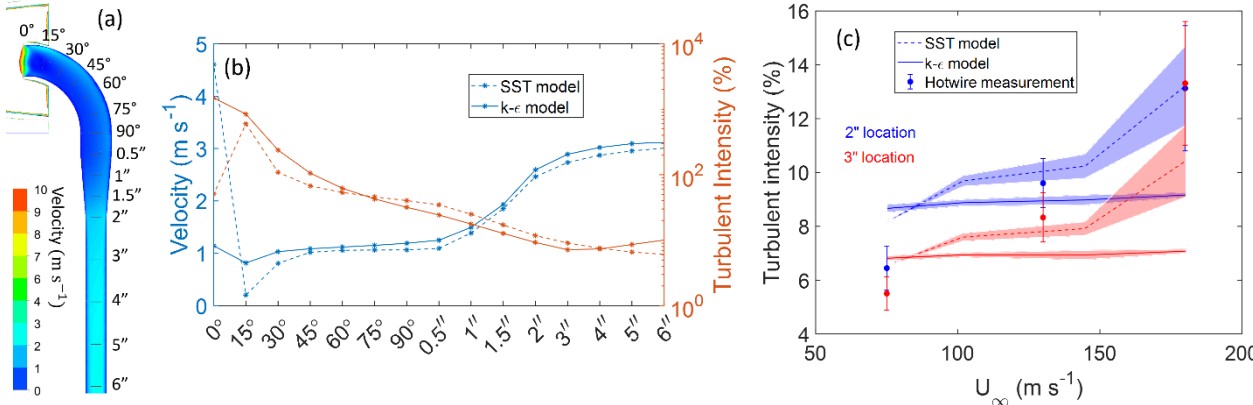

Figure 4: Simulation results inside sampling tube for ground level conditions in Table 1 (180 m s$^{-1}$, 25mm restrictor, 3% freestream turbulence, 3° angle of attack). (a) Contour of velocity magnitudes in the center plane of the sampling tube, with different locations relevant for (b) labelled. (b) Surface weighted average of turbulent intensities and velocities along the length of the sampling tube. (c) Comparisons of turbulent intensity measured and simulated inside the sampling tube. The simulated sampling flow rate is Q$_{sampling}$=~19SLPM; the shading reflects varying TI$_\infty$ from 0.5% to 3% in the models. The hotwire measurements show average
values for 15 and 20 SLPM sampling flowrate; error bars represent the range of measurements at the two sampling flow rates.

A comparison of the CFD results with experimental measurements of turbulent intensities inside the sampling tube obtained from hotwire measurements is shown in Figure 4c. The uncertainty in experimental data corresponds to the range of turbulent intensity measurements for sampling flow rates from between 15 SLPM to 20 SLPM. The experimental data confirms that the turbulence intensities are dampened as the flow travels through a greater distance in the sampling tube. The turbulent

intensities at both locations in the sampling tube increase with increasing freestream velocity, consistent in the trend seen with SST model and different from that predicted by k-ε model. Quantitatively, the experimental data did not exactly match the observed intensities. But considering the uncertainty in the freestream turbulence in the wind-tunnel and hot-wire measurements, the experimental data can be assumed to have validated numerical results of SST model.

**4 Discussion:**

Due to ease of convergence and reliable mean flow results, the k-ε turbulent model is widely used to model flow fields of aircraft-based sampler, especially aerosol sampling inlets. While the average flow velocity is identical for the two turbulence models studied, k-ε and transition SST, the wind-tunnel measurements of turbulent intensity suggest that only the transition SST model reasonably predicts inlet flow fields under aircraft sampling conditions. This is a critical finding, providing confidence in the use of RANS simulations for inlet performance characterization and design optimization under high-altitude

aircraft operating conditions.

CFD simulations using the validated SST model were used to optimize the inlet design for maximal gas sampling efficiency under high-altitude conditions. At high-altitude conditions, for inner shroud exit restrictor sizes varied from 25 mm to 6.25 mm, the velocities and turbulence intensities at location A (Fig. 1b) are shown as solid lines in Figure 5. Decreasing the restrictor size decreased the inner shroud velocities from ~ 35 to 2 m s$^{-1}$ while correspondingly increasing the turbulent intensities at the location A from 10% to ~ 100%. The inner shroud velocity and turbulence intensity determine flow turbulence in the sample tube and in turn the efficiency of gas/particle transport in the entrance region of the sampling tube. Minimizing turbulence in the sampling tube requires optimally balancing the turbulence in the flow upstream of the tube and relative velocity of the upstream flow to that in the sample tube. For the different restrictor sizes, the turbulence just inside the sampling tube at the 15° location in the bend (Fig. 4a) is shown in the same Figure 5. With decreasing the size of the restrictor is seen to decrease the turbulence inside the sampling tube entrance. A similar trend is observed for ground-level conditions for three restrictor sizes with 180 m s$^{-1}$ freestream velocity (Dash lines in Fig. 5).

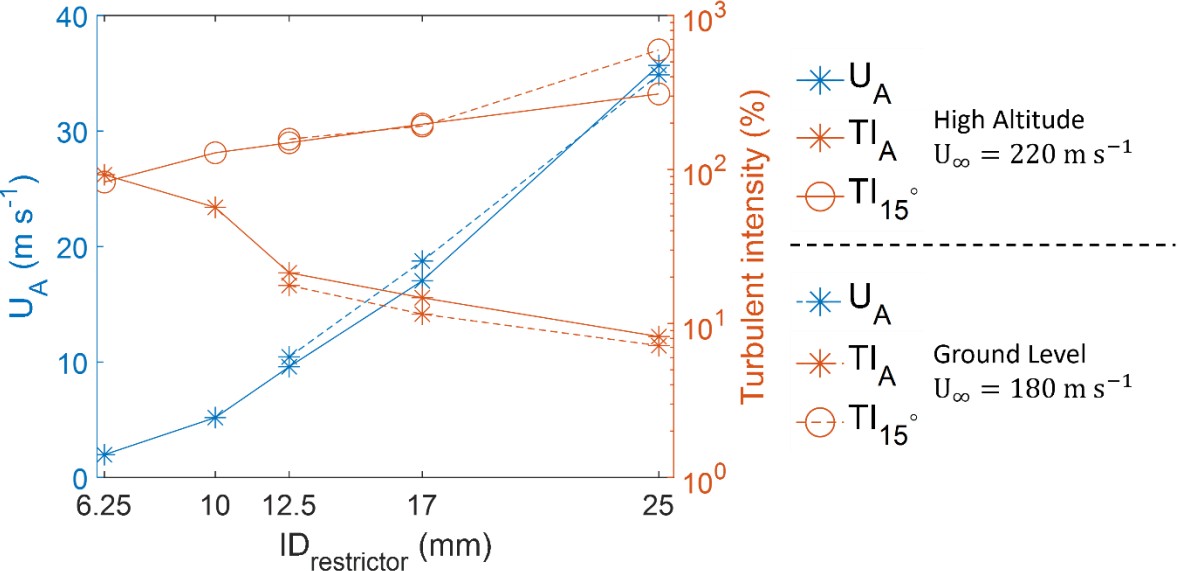

**Figure 5: Simulation results of different sizes of restrictor at sampling flow velocity 2.4 m s$^{-1}$. The left y-axis shows velocity ($U_A$) at location A, the right y-axis shows turbulence intensity ($TI_A$) at location A and turbulent intensity ($TI_{15°}$) at location inside sampling tube at 15° bend. Solid lines represent high-altitude freestream conditions for a freestream velocity of 220 m s$^{-1}$ and 15,000 Pa; dash lines represent ground level freestream conditions for a freestream velocity of 180 m s$^{-1}$ and 97,000 Pa. See Figure 1b and Figure 4a for location information.**

To minimize the net turbulence in the inlet, an optimal restrictor size is ~ 12.5 – 17 mm. Smaller sizes (<12.5 mm) would increase turbulence in the inner shroud and thus significantly increasing gas loss upstream of the sampling tube. Larger restrictor sizes will increase flow velocities in the inner shroud and enhance turbulence in the sampling tube where flow is slowed down. This increases turbulence in the sample tube and hence likely to increase gas losses inside the tube.

To quantify the impact of turbulence on gas transport through the sample tube, gas species mass transport equations must be

integrated with the CFD flow simulations. Here, we consider water vapor transport through the sample tube with perennially dry walls (i.e., the walls are a perfect sink, with a zero water vapor concentration at all times) and determine loss of the vapor to the wall. The gas-phase transport efficiency is defined as the mass fraction of water at any cross-section compared to a reference mass fraction of water. In addition, using the mass fraction of water vapor at the sampling outlet divided by the freestream mass fraction of water vapor, we calculated the overall gas sampling efficiency (transmission) under different

sampling flow rates (5 to 40 SLPM), restrictor sizes, and freestream velocities under both high-altitude and ground level cases. The simulations results (Fig. 6a) show that the vapor is primarily lost after entering the sampling tube, with the k-ε model predicting 10~20% higher overall gas loss compared to the SST model due to its higher prediction of turbulence diffusivity (Fig. S3). The overall gas sampling efficiency is seen to be most highly correlated with sampling flow rate over the other parameters. An exponential correlation explains the relation between residence times inside the sampling tube and overall

sampling efficiency for the gas exiting the sampling outlet (Fig. 6b). This observation suggests that optimal sampling operation requires maximizing the sampling flow rate rather than restricting the flow to laminar. This is an important aircraft gas-sampling recommendation that will need to be validated with appropriate gas transport measurements under controlled wind tunnel conditions. However, inlet transmission is only one criteria under which to optimize sampling flow, and the instruments sampling downstream of the inlet set additional constraints on the choice of laminar versus turbulent flow is desirable.

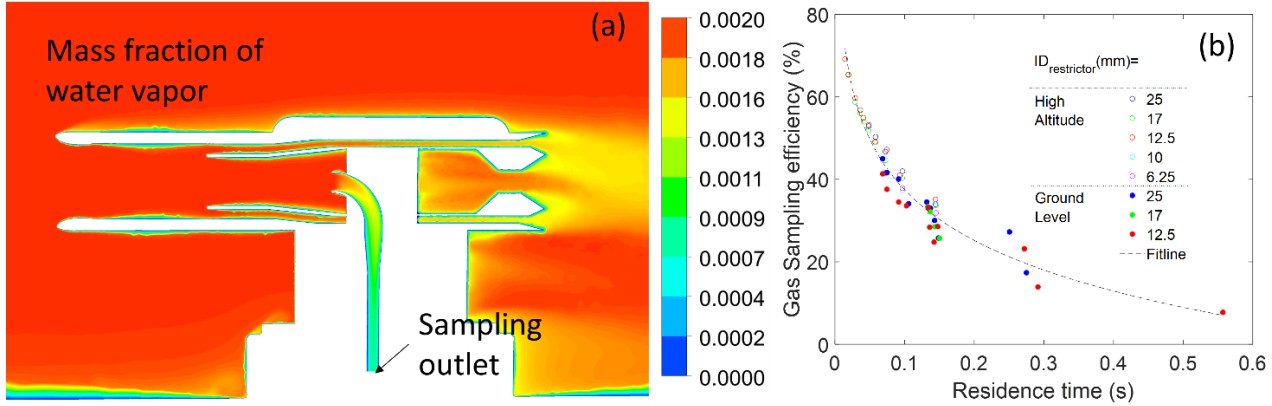


Figure 6: (a) contour plot of mass fraction of water vapor of the inlet for 25mm restrictor with freestream velocity of 180m s⁻¹, freestream turbulence intensity of 3%, and angle of attack of 3°. (b) correlation between residence time inside the sampling tube and the overall gas sampling efficiency at the sampling outlet for all restrictors under different altitudes, freestream velocities and sampling velocities.

To optimize the design and operation of the gas inlet, it is important to understand the relation between the factors driving the loss of gas-phase species during transport and the inlet sampling conditions and design features. For this, we split the analysis of the inlet sampling characteristics into two sections – the entrance section that extends through the initial 90 degree turn to the 2" location of the sampling tube; and the downstream section, which extends from the 2" location to the end of the tube (sampling outlet in Fig. 6a). The entrance section is seen to have the highest turbulent intensity in the sampling tube and

correspondingly will experience a higher rate of gas loss rate than the rest of the tube. For the different cases studied, the

transport efficiency of gas-phase species through the entrance section is seen to be inversely proportional to the turbulence intensity in the entrance region of the sampling tube ($TI_{15°}$) (Fig. 7a). The turbulence intensity in the entrance section reduces with increasing relative sampling velocity, i.e. the ratio of sampling velocity ($U_{sampling}$) to that just upstream of the sampling tube entrance ($U_A$). As the relative sampling velocity approaches one, i.e. when near isokinetic sampling is established, the

sampling tube entrance turbulence intensity, $TI_{15°}$, is minimized. Thus, an ideal gas inlet design will have a restrictor that will reduce the upstream velocity to closely match the sampling velocity. To minimize gas loss in the entrance section, the restrictor size selection must also consider turbulence intensity in the upstream flow ($TI_A$). As the restrictor size is reduced to decrease velocity $U_A$, the turbulence intensity in that section increases, resulting in upstream loss (Fig. 7b).

In the downstream section (from 2" location to outlet), however, the gas transport efficiency is independent of the upstream conditions, and dependent only on gas residence time in the tube (Fig. 8). At the lowest sampling velocity, the efficiency will be limited by laminar diffusion. Increasing the sampling velocity will result in flow turbulence and an efficiency below the laminar limit. Even under turbulence, increasing the sampling velocity results in increasing the sampling efficiency.

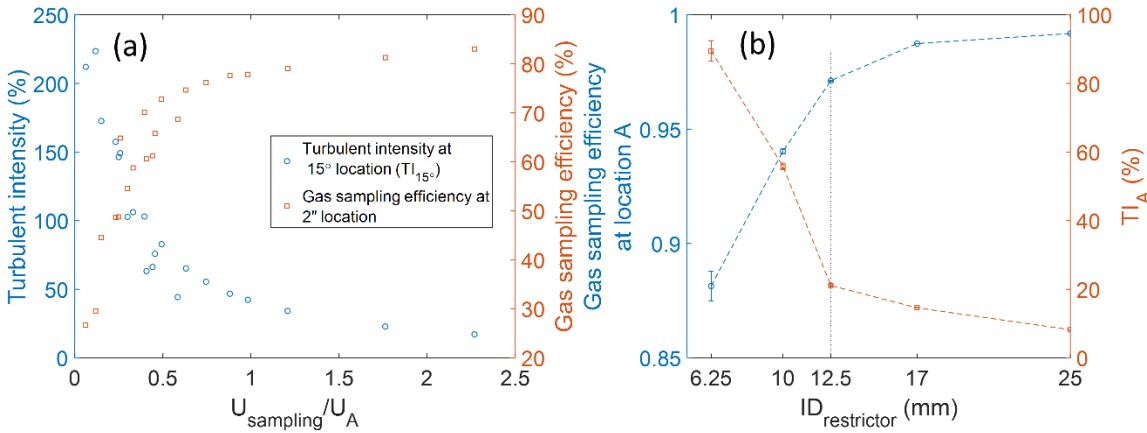


**Figure 7: (a) Simulation results of 12.5mm restrictor from all conditions at different relative sampling velocity ($U_{sampling}/U_A$). The left y-axis shows turbulent intensity ($TI_{15°}$) at location inside sampling tube at 15° bend, the right y-axis shows the gas sampling efficiency after the bending area (at 2" location). (b) Simulation results of gas sampling efficiency and turbulent intensity at location A with different sizes of restrictor at high-altitude freestream conditions for a freestream velocity of 220 m s$^{-1}$ and 15,000 Pa. The**
**uncertainty bar in this plot represents the standard deviation from different sampling velocity cases.**

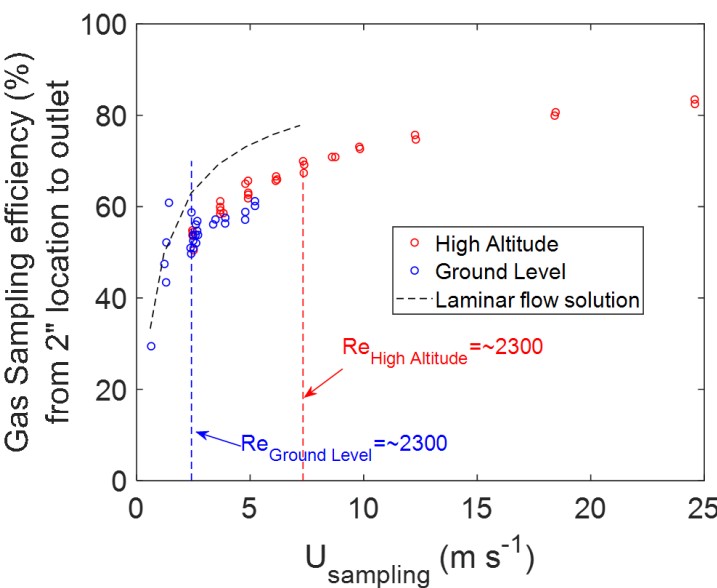

**Figure 8: Correlation between gas sampling efficiency in the straight tube (from 2" location to outlet) and the sampling flow velocity for all cases. As a reference, the black dash line is the exact solution by solving the mass transport model under a 2D-axisymmetric laminar pipe flow with constant uniform velocity and diffusivity of water vaper.**

The diffusion coefficient in our model accounts for both laminar diffusivity and turbulent diffusivity. Laminar diffusivity is determined by the gas species, while turbulent diffusivity is calculated based on the turbulent viscosity predicted from the flow model. In turbulent flow, turbulent diffusion will generally overwhelm laminar diffusion, rendering the latter less important in most of the flow domain. In our study cases, turbulence dominates at the entrance and then dissipates along the sampling tube. Thus, the dominance of turbulent diffusivity diminishes (Fig. S4) along the flow length and the laminar diffusivity value

becomes subsequently important, particularly near the sampling outlet. For $H_2SO_4$, the overall sampling efficiency at the sampling outlet, shows a 5%~20% lower loss compared to water vapor (Fig S3).

As the model of mass diffusion loss is concentration gradient-dependent, the overall gas loss at the sampling outlet exhibits a linear relationship with the ratio between the mass fraction of species at the wall and in the flow (Fig. S5). Gas loss occurs

when the mass fraction at the wall is less than the mass fraction in the flow, and no gas loss occurs when they are equal. If the ratio between the mass fraction at the wall and in the flow is greater than 1, species will migrate from the wall into the flow, contaminating the sample flow. In this paper, we only focused on the case of the wall as a perfect sink, representing the worst-case scenario for gas transport loss.

This manuscript focuses on the description and characterization of fluid dynamics by measurements and simulations. Initial attempts to measure the chemical transmission inside the windtunnel using $H_2SO_4$ as described above have yielded mixed results; due to low $H_2SO_4$ signal most likely related to low photon flux from the light source, impurities in the windtunnel air,

or a combination of these effects. Additional windtunnel time has been requested, but no measurements of chemical transmission are available at this time.

## 5 Summary

A gas-inlet design based on a forward sampling probe is studied using CFD simulations and wind-tunnel experiments to establish its sampling performance. In this inlet design, the flow velocity gradients through the inlet can be varied by varying the exit restrictor section. The turbulent interaction of sample gas with walls is minimized to maximize transport efficiency of condensable vapor, and the inlet is designed such that the freestream velocity is smoothly reduced from the high aircraft speed to a much lower velocity just upstream of the sample tube, using two shrouds.

CFD simulations show that the calculated turbulence intensities in the inlet depend on the choice of the turbulence model. The SST model predicts lower turbulence intensities in the shroud than the k-ε model. Both turbulence models predict a similar general trend of turbulence intensities along the length of the flow inside the sampling tube. However, the SST model predicts lower turbulence intensities that vary with freestream turbulence and significantly increase with freestream velocities. While the k-ε model predicts larger turbulence intensities that are largely invariant with freestream conditions. The very different predictions of the two turbulence models needed to be resolved before further CFD simulations could be undertaken to optimize the inlet design for sampling performance.

Flow experiments using pitot-tube and hotwire measurements inside the inlet were conducted using a full-scale inlet placed in a high-speed wind-tunnel. The inner shroud mean flow velocity was measured using the pitot-tube, while hotwire measurements were made at several locations within the inlet to determine local velocities and turbulence intensities as a function of freestream conditions. The inner shroud mean flow velocity variation with sizes of restrictor, angles of attack, and freestream velocities were seen to be in good agreement with simulation results. Hotwire measurements of turbulent intensities in the inner shroud and sampling tube for varying wind tunnel flow velocities show a good agreement with the transition SST model, validating the use of this model for further flow calculations and inlet design optimization. Using the validated CFD turbulence model, it was determined that a restrictor size in the range of 12.5 to 17 mm diameter allowed for optimal sampling conditions that minimized the net turbulence intensity in the inlet.

Preliminary calculation of gas transport efficiencies in the sample tube suggests that with the inlet entrance turbulence, it is necessary to maximize the flow rate in the sample tube to minimize transport loss, rather than slow the flow down to laminarize the flow. The specific conclusions on what type of flow is desirable will depend on the instrument configuration and application.


**Author contribution:**

DY, RV, RM and SD designed the experiments and calculations. DY and SD developed CFD model and conducted the simulations. MR, RM, RV and SD carried the wind tunnel experiments. DY prepared hotwire data acquisition system, calibrated the instrument, and analysed the data. DY, RV and SD wrote the manuscript, with contributions from all coauthors.


**Competing interests:**

At least one of the (co-)authors is a member of the editorial board of Atmospheric Measurement Techniques.

**Acknowledgment**

Financial support from the National Science Foundation is gratefully acknowledge for the initial inlet design (EAGER award AGS-2023961), and windtunnel tests (AGS-2027252, 2027262). The authors thank Kenneth Smith from the Integrated Instrument Development Facility at CIRES for help with fabrication, and Prakriti Sardana for initial help with the wind tunnel tests. The wind tunnel experiments were conducted at the US Air Force Academy Aeronautics Research Center under Commercial Test Agreement 21-161-AFA-01.

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
