# Peer review of "Performance characterization of a laminar gas-inlet"

_Atmospheric Measurement Techniques, 2023_

## Author Response (AR2)

**Comment 1 (RC1)**

The authors thank the reviewer for their comments.  Please find our responses below.

Comment:

The authors present results about the characterization of an inlet for sampling air into an aircraft. As the authors state, the focus of characterizing such inlets has been on the sampling of particles and less on gas-phase species. In this paper, the authors focus on the characterization of turbulence of a forward-facing laminar flow gas inlet that is an improved version of a previously used inlet. CFD modelling is compared with some measurements in a wind tunnel showing that some parameters are better described by CFD modelling using the shear tress transport model. There is little to no experimental characterization of the gas-phase inlet concerning the transmission of gas-phase species, which would be of high interest for the atmospheric community and what I had expected to see after reading the abstract.

Therefore, this paper is mainly describing the engineering aspects of CFD modelling and the model results accompanied with wind tunnel experiments. It is not very clear if there are results that can be generalized or if results only apply for the specific inlet described in this work. As non-expert on CFD modelling and the descriptive character of the paper, I cannot judge, if the modelling on its own is worth being published in AMT. Overall, the paper reads to me like an engineering report that is certainly needed during the development of such an inlet rather than a research paper in atmospheric sciences.

Response:

We appreciate the reviewer's comments on the paper.   The reviewer correctly points out that the paper is focused on engineering evaluation of a gas inlet using CFD and wind-tunnel measurements.  We, however, disagree with the reviewer's comment about the extent of applicability of the paper to the AMT audience.  A reading of the mission statement of AMT listed below (relevant words highlighted in bold) clearly suggests that the objectives of this paper fits perfectly with the mission of AMT.

The mission statement from the AMT journal website:

"The main subject areas comprise the **development**, intercomparison, and validation of **measurement** instruments and **techniques** of data processing and information retrieval **for gases**, aerosols, and clouds. Papers submitted to AMT must contain atmospheric measurements, **laboratory measurements relevant for atmospheric science**, and/or **theoretical calculations of measurements simulations** with

detailed error analysis including **instrument simulations**. The manuscript types considered for peer-reviewed publication are research articles, review articles, and commentaries."

Aircraft inlet studies are commonly published in AMT (e.g. Sanchez-Marroquin, et al., Atmos. Meas. Tech., 12, 5741–5763, 2019) and AMT publications of such studies are dependent on CFD simulations (e.g. Moharreri, A., et al. Atmospheric Measurement Techniques, 2014). Thus, related work on evaluation of aircraft inlets in wind-tunnels would be very relevant to the AMT audience. More importantly, the role of turbulence on transport of particles and gases in aircraft inlets is known to be important but often ignored by the community (including in the papers above) because of the challenges of getting turbulence right. This paper provides important guidance on that front – demonstrating the significant differences between model approaches, and the accuracy of certain CFD models over others (unfortunately the commonly used models are often less accurate, as seen here, but more widely used because they more easily converge). While this paper could be sent to a fluids-related journal, our choice in publishing this in AMT is driven by the need to have validated modeling approaches relevant to the atmospheric community be visible in a journal relevant to the community.

Specific comments:

Introduction: It is not clear what the authors mean with "gas-phase transport efficiency" mentioned in line 44. There should be a clear definition.

Response:

The gas-phase transport efficiency is defined as the mass fraction of water at any cross-section compared to the ambient mass fraction of water. This is clarified in the paper now.

Discussion: I would have expected to read much more about the consequences of the results concerning air sampling / loss of species and experimental characterization. In my opinion this would be needed for a paper in AMT. There is no discussion with results reported in literature as gas-phase sampling using such inlets have been applied in numerous previous aircraft campaigns.

Response:

This paper uses the validated CFD model to demonstrate two important findings: 1) the model challenges the very widely made assumption in the atmospheric community that laminar flow in inlet lines, and core sampling from such lines is preferable over turbulent sampling; and 2) the use of an incorrect model (widely used k-ε model) results in prediction of 10-20% higher losses than predicted by the validated k-ω SST model. These findings are described in pages 13 and Figure 6b of the original paper. These findings support that the benefit of minimizing the residence time by accepting turbulence far outweigh attempts to minimize wall losses by laminar core sampling. We will expand on this aspect in the revised manuscript.

We are in the process of finishing experiments to fully characterize sampling losses of species in the inlet under high-speed wind-tunnel conditions. As might be expected, wind-tunnel measurements of gas transport efficiencies are quite challenging and that work will merit its own paper. The current paper stands on its own merit, as the validated calculation of turbulence characteristics in the inlet is relevant for any gas and aerosol sampling. About results of gas-sampling inlet efficiency in the literature, we are unaware of such published studies that we could take advantage of for our validation. We would greatly appreciate any pointers in this regard.

Line 33: "the" is missing between "inside" and "aircraft"

Response: The typo is now fixed.

Line 265: "understanding" instead of "understand"

Response: The typo is now fixed.

We appreciate the opportunity to respond and look forward to any additional comments there might be.

**Comment 2 (RC2)**

The authors thank the reviewer for their comments. Please find our responses below.

Comment:

Yang and coauthors describe a novel inlet for measuring gas-phase species on aircraft platforms. Using a computational fluid dynamics (CFD) model, they simulate the behavior of sampled air inside this inlet, and complement their simulations with

observations taken of a prototype inlet inside a high-speed wind tunnel. They report flow speeds and turbulence intensities for several combinations of inlet and sampling parameters. Finally, they estimate the throughput efficiency of the inlet.

This is a well written report that describes an inlet that often isn't characterized as well as aerosol-phase inlets. I believe it is a valuable contribution to the field, and I would recommend publication, following some minor modifications.

Response:

Thank you very much for your comments and suggestions below.

Comment:

**General**: The authors have spent considerable time engineering an inlet that maintains as close to laminar flow as possible, by reducing turbulence. However, their simulations in Section 4 indicate that laminar flow might not necessarily be the factor that reduces losses in the inlet. The authors should spend more time exploring this finding, as I'm concerned it partially undercuts their results. If laminar flow isn't the key factor in reducing losses, then is this really the ideal inlet configuration for gas-phase sampling?

Response:

The reviewer accurately notes that the inlet was engineered to provide laminar flow for maximal gas inlet transmission. As correctly summarized, one of our critical findings is that minimizing the residence time is more important than maintaining laminar flow. At any selected Reynolds number, turbulence in the transportation tube will lower the transmission efficiency relative to laminar flow. However, even considering turbulence, decreasing the residence time sufficiently can enable higher transmission efficiency than obtained at the limit of laminar flow regime.

In addition to the transportation efficiency over the length of the sample tube, the gas sampling efficiency also depends on the loss mechanisms acting at the sampling tube entrance. Our simulations show that the turbulence in the flow just downstream of sampling tube entrance ($TI_{15°}$) is highly influenced by the ratio of the sampling velocity to that just upstream of the sampling tube entrance ($U_A$). Additionally, the sample flow turbulence at the entrance ($TI_{15°}$) is also a function of the turbulence intensity just upstream of the sampling tube ($TI_A$) under the same flight condition. Our simulations show that minimizing the upstream turbulence intensity and maximizing the ratio of sample velocity to upstream velocity by appropriate selection of restrictor size, maximizes the transmission efficiency of the sampled gas through the entrance region.

As recommended, we have updated the text to discuss these findings in greater detail than done previously. Considering the absence of literature on gas inlet design guidelines, adding additional details of our findings will improve the impact of our paper over time and we appreciate the reviewer's advice on this.

Comment:

Some additional calculations may be helpful which estimate losses when a given chemical species has a theoretical loss probability that is less than 100% upon collision with the wall. Reading that section, I think the authors are assuming that as soon as a molecule collides with the wall then it is lost. In that case, another water molecule will come take its place, and will also be lost when it collides with the wall surface. However, if the loss probability is less than 100%, then wouldn't a laminar flow will have a very thin layer of molecules that are repeatedly interacting with the walls, but not necessarily being lost? I would expect that this simulation would reduce the estimated losses, and would be more physically reasonable for most chemical species.

Response:

The reviewer is correct that the transmission efficiency results depend on the wall boundary conditions for the gas species. If the accommodation coefficient is less than 1 then gas loss to walls is reduced. The currently presented results are based on the assumption of a perfect accommodation of gas species coming in contact with the wall and represents the "worst-case" scenario for transmission loss. We have now included in the supplemental material, the calculated species loss when the accommodation coefficient on the wall is less than 1.

As noted by the reviewer, the boundary layer flow is likely to include a laminar sub-layer that will act to limit the extent of wall loss, relative to the case of a fully turbulent boundary layer. Using SST turbulence flow modeling results in accounting for the effect of the laminar sub-layer in the turbulent boundary layer moderating the species wall loss.

Another factor of importance for transmission loss calculation is the gas diffusivity coefficient. Considering the high diffusivity of water vapor molecules, the presented calculation represents one of the "worst-case" scenarios for our calculations. For a heavier molecule like $H_2SO_4$, the transmission loss will be lower (about 5 to 20% lower for the cases studied) than that for water vapor and this information with new simulation results are included in the revised Supplemental material.

Specific comments:

Line 12: Replace "|" with ", and"

Response:

The typo is now fixed.

Line 68: "Using elliptical cross-sections for the leading edge..." is unclear. Which dimension is elliptical in Fig 1?

Response:

The inner and outer shroud leading edges were described in detail as 10cm and 5cm ducts from "Chapter 2. Design Criteria" of paper "An inlet sampling duct for airborne OH and sulfuric acid measurements" (Eisele et al. 1997)

Line 70: Can you describe in more detail what "flow straightening" means in this context. How do you quantify this?

Response:

The "flow straightening" we describe here means that the flow passes through the blunt body smoothly without generating any recirculating flow or flow separations. We quantify this by investigating the streamline or path line surrounding the entrances of outer and inner shroud. This is in section 2.1 paragraph 1.

Section 2.1: You describe in detail how a chemical calibration has been done in the past, leading me to wonder why it wasn't done in this paper. Why is it "beyond the scope of this paper"?

Response:

Chemical calibration of aircraft inlets under high-speed wind-tunnel conditions are complicated by challenges of temporal variance of generated species under different flow conditions, low detection levels, and setup of mass spectrometer for real-time measurements. In response to the reviewer comment, we now briefly summarize these challenges in the revised paper, arriving at why it is beyond the scope of this paper.

We have added the following text to the revised manuscript: "This manuscript focuses on the description and characterization of fluid dynamics by measurements and simulations. Initial attempts to measure the chemical transmission inside the

windtunnel using $H_2SO_4$ as described above have yielded mixed results; due to low $H_2SO_4$ signal most likely related to low photon flux from the light source, impurities in the windtunnel air, or a combination of these effects. Additional windtunnel time has been requested, but no measurements of chemical transmission are available at this time. " In the revised paper, we have included a discussion of additional simulations involving the diffusivity of $H_2SO_4$ within the current flow domain.

Line 217: What is the basis for the 0.5 – 3% range? Is that based on physical parameters or is that the range needed to span the observed turbulence intensities?

Response:

The wind-tunnel free stream conditions of turbulent intensity for simulation were experimentally determined from velocity measurements made using pitot tubes in the wind tunnel freestream flow. The wind tunnel velocity measured by pitot tube is recorded very 10s. Calculating the velocity fluctuation from wind tunnel operating data, we observed that the estimated intensity of wind tunnel at different free stream conditions is at the 0.5%~2.8% range. Combining the empirical estimations, we conducted the simulations under 0.5%, 1% and 3% free stream turbulent intensity respectively.  This info is now added to the supplement.